

# Analysis of applicability of flood vulnerability index in Pre-Saharan region, a pilot study to assess flood in Southern Morocco

A. Karmaoui[1] , S.F. Balica[2] , M. Messouli[1]

[1] LHEA (URAC 33). Department of Environmental Sciences, Faculty of Sciences Semlalia, Cadi Ayyad University, Marrakesh, Morocco E-mail: Karmaoui.ahmed@gmail.com
[2] E-mail: s.f.balica@gmail.com.
[1] LHEA (URAC 33). Department of Environmental Sciences, Faculty of Sciences Semlalia, Cadi Ayyad University, Marrakesh, Morocco E-mail: messouli@gmail.com

*Correspondence to*: Stefania Balica (s.f.balica@gmail.com)

**Abstract.** Moroccan Pre-Saharan zone is an oasis system, which it is characterised by extreme events, like drought and flood. The flood risks will likely increases in frequency and magnitude due to global and regional climate change. Flood tends to have an important impact on isolated and poor regions such as oasis regions. This paper aims the analysis of applicability of Flood Vulnerability Index (FVI) in pre-Saharan region of Morocco. The FVI, it is a numerical index that reflects the status of a region's flood vulnerability. It was determined for four components social, economic, physical, and environmental. These components can help to assist to propose strategies for improvement of the holistic system. For this study five sub-catchments were selected: Upper Draa Valley (UDV), Middle Draa Valley (MDV), Tata sub-catchment, Guelmim sub-catchment and Tafilalt sub-catchment; and five urban areas, Ouarzazate, Zagora, Tata, Guelmim and Errachidia. A comparative analysis of the results from thus areas allows us to assess the applicability of the FVI. The overall FVI for these areas was determined by the calculating and standardisation of 36 indicators for each sub-catchment scale and 34 for each urban scale.

Keywords: Flood, Vulnerability, oasis, environmental impact, climate change, adaptation

## 1. Introduction

Globally, dry lands areas is estimated to be about 41 percent of the terrestrial surface, and are home to a third of humanity, and concentrate the high rate of poverty (Mortimore *et al,* 2009). Dry lands are located mainly in poor countries: 72% of this area is found within developing countries and only 28% within industrial ones (MEA 2005). Morocco is one of these countries. In fact, is located in the North-West corner of Africa, bordered by the Mediterranean Sea and the Atlantic Ocean on the North and West, by Algeria on the East, and by Mauritania on the South. Its total land area is 710850 km$^2$ and includes different landforms, like agricultural plains, River valleys, plateaus, and mountain chains (Anon 2004). Most of these lands are arid to semi-arid from which 75% are rangelands, 13% forests and 8% are cultivated (Dahan 2012). In the hyper-arid and arid dry lands (the desert biome), most agricultural activities are in oasis, where the irrigation is by fluvial, ground, or local water sources (MEA 2005). The drought threatens severely the populations in the study areas. In fact, a large part of the date palm trees (main source of life) in the Southern zone died, causing a reduction from 4575 km$^2$ to 1342 km$^2$ (Benmohammadi 2000). Furthermore, as palm trees are the main element for the oasis agriculture, the number of animals also decreased because of the drought event (Heidecke & Roth 2008, Ait Hamza *et al.* 2009).This dependence of oases on water makes this area, highly vulnerable to extreme events, like droughts and floods.



Climate change causes acceleration in the frequency of extreme events. Human societies have developed in trying to cope by limiting impacts. In this context the IPCC (2012) has developed risk management strategies. According to the IPCC report, the impacts of changes in flood are highly dependent on how climate changes in the future (IPCC 2012). That is happening for droughts as well. In fact, the impact of natural disasters is correlated to the vulnerability of communities in developing countries, as previous socio-economic vulnerabilities may accelerate these disasters, making the recovery very difficult (Vatsa & Krimgold 2000). Thus, the impact of such events increases the poverty (Carter *et al.* 2007). Historically, flood has damaged properties infrastructure and thousands of populations. In Morocco, floods are the most dangerous natural disasters, as seen in Fig. 1 and Table 1. The number of affected people and lives lost due to floods exceeds any other natural disasters in the past thirty years. The data related to human and economic losses) from disasters that have occurred between 1980 and 2010 in Morocco, according to UNISDR (UN Office for Disaster Risk Reduction (www.preventionweb.net), can be seen on Table 2). During this period (1980-2010), 78 dams have been built at the national scale.

Figure 1: Natural Disaster Occurrence. Source: UNISDR (UN Office for Disaster Risk Reduction)

Table 1: Total number of people affected since 1963 due to flood in Morocco. Source: "EM-DAT: The OFDA/CRED International Disaster Database. www.em-dat.net"

Table 2: Human and economic losses from disasters occurred between 1980- 2010, in Morocco

In recent decades, Morocco has experienced several extreme events. These hydro-meteorological events have greatly impacted the economies. Floods are leading these events. Indeed floods caused significant damage to the socio-economic side. In this context, can be cited a few examples of recent major floods occurred in Morocco in general and their socio-economic consequences:

- On 25[th] of September 1950 a flash flood of 6m of height flooded the city of *Sefrou* making a hundred victims (Saidi *et al.* 2010):

- On 23[rd] of May 1963 a violent flood of 7200 m³/s of peak flow devastated the Moulouya valley taking the left seat bank of the dam Mohammed V, according the official website of the Moroccan Ministry of water (www.water.gov.ma).

- Finally, the famous flood that affected watersheds Marrakech High Atlas on 17[th] of August 1995. In this region flood of about 1030 m³/s occurs (RIAD 2003).This flood made 730 victims and 35 000 affected (CSIG 2008).

During the last decade, the notion of vulnerability has changed. After IPCC in 2001 hazards such as climate change, defines vulnerability as "the degree to which a system is susceptible to or unable to cope with, adverse effects of climate change, including climate variability and extremes". Connor & Hiroki (2005) developed a Flood Vulnerability Index (FVI), which allows for a comparative analysis of flood vulnerability between different River basins. Methodology which let operators to recognize the key causes conscience-stricken for the basin's vulnerability. Vulnerability is expected to happen under certain conditions of exposure, susceptibility and resilience, measuring the extent of harm (Fuchs et al. 2011). The present article will use the following definition of vulnerability specifically related to flooding (Balica *et al.* 2009): the degree to which a natural or man-made system is susceptible to floods due to exposure, a perturbation, in co-occurrence with its ability (or inability) to



cope, recover, or basically adapt. Managing risks from floods should be an important component of climate change adaptation. This study focuses on an approach to assess flood vulnerability and discuss the benefits of adaptation options at a city-scale and sub-catchment-scale, using Flood Vulnerability Index (Balica *et al.* 2009) in pre-Saharan region. The indicators used show the variables affecting the flood vulnerability in the pre-Saharan region. It provides an important tool for decision maker for monitoring and evaluating changes over time. Data was collected from a variety of sources (see Appendices 2), including household surveys, documents, government and ministries. The data gathered pertained to the particular indicator to be calculated. The FVI is an indicator-based index which reflects the status of a scale's flood vulnerability. This index was determined for four components social, economic, physical, and environmental. As indicated in the title of this article, the paper aims to assess flood vulnerability in Southern Morocco; in oasis basins, by analyzing the applicability of FVI. Five sub-catchments and five urban areas were selected; the results from thus areas allow us to assess the applicability of the FVI.

## 2. Materials and methods

### 2.1 Study area

Pre-Saharan North Africa constitutes a major indicator of climatic trends in the Mediterranean region; is currently experiencing a rapid climatic deterioration and desertification (RBOSM 2008). This situation makes the region (see Fig. 2) a vulnerable area. In fact, since the middle of the twentieth century, oases have borne increasing demographic and investment pressures resulting in larger water abstraction, soil salinization, loss of surrounding vegetation and soil erosion (MEA 2005).

Figure 2: Moroccan pre-Sahara: Oasean zone, including the basins of Guelmim, Tata, Zagora (MDV), Ouarzazate (UDV) and Errachidia (Tafilalet)

The study area corresponds to the perimeters of the oasean provinces of Zagora, Ouarzazate, Guelmim, Tata and Errachidia. It is located in part of the Draa Basin (Upper, Middle and low Draa), and of Tafilalet basin. At the economic level of the entire area, agriculture occupies a prominent place in the economy of these provinces; in fact it is one of the main sources of income and occupies the major part of the workforce (EVICC 2011a). Industrial activity is almost non-existent and the tourism activity remains well below the existing potential. These oasis basins are located near the Wadis (temporary rivers) to facilitate the use of water surface. This location near the Wadi beds is important for the mobilization of water, however it causes the exposure of these areas to flood risk (EVICC 2011e). The floods are rare in the basin of Draa (UDV, MDV, Tata sub-catchment and Guelmim Sub-catchment) but they are brutal and violent (PACC 2012a).

The study area has experienced several floods, causing considerable socio-economic damages; we will quote the most important in the study area:

### 2.1.1 Tafilalet

On the 5th of November 1965, a flood destroyed the Ziz valley (Tafilalet), leaving 25,000 people homeless and accelerating the decision to build the dam Hassan Addakhil (Saidi *et al.* 2010). In Merzouga, the last important





flood was recorded on the 26th of May 2006 after an intense rainfall (112 mm/3 hours) (Minoia & Kaakinen, 2012). The flood damages were significant, with the destruction of 140 houses and hotels, deterioration of Taouz–Merzouga road and of the ONEP (National Agency for drinking water and sanitation) water supply pipe of Merzouga villages (See Fig. 3, Photos: A and B) and Taouz (Kabiri 2012).

Figure 3: A: Collapsed houses in Merzouga.  B: State of the road after a flood, at the entrance of Merzouga (Source: Kabiri, 2012).

It was happening the same on the Rheris River (Tafilalet) where in 1965 an observed and measured average annual rate of 9.2$m^3$/s took place (PACC 2012a).

The same phenomenon was observed on the Dadès River(Upper Draa Valley): between 1965-1966 the Dadès Riverand its tributary Assif Mgoun, respectively recorded average discharge of 7.8 $m^3$/s and 12.3 $m^3$/s with an annual contribution of 103.6 $m^3$/s and 147 $m^3$ in the same order.

### 2.1.2 Draa (UDV sub-catchment and MDV)

The drainage system of Upper Draa consists by temporary RiverOuarzazate and Douchen in the Westside and by perennial Oued Dades Mgoun in the East side. They are fed by karstic aquifers that originate in the high mountains of the north east of the High Atlas. This aquifer is fed by melting snow and water infiltration. Upper Draa receives an average of 514 M$m^3$ (EVICC 2011d).The drainage system of the Middle Draa is less densed and drained mainly by the Draa and its tributaries. The average flow recorded in Zagora is 13.4$m^3$/s and the maximum that is ever recorded it is in 1965 and reached 213$m^3$/s. However, in August of the same year, this zone knows only 0.13$m^3$/s (Ait-Hamza *et al.* 2009). The violence of flood causes the phenomenon of water erosion which reduces the fertility of agricultural land (EVICC 2011d).

The photos (See Fig. 4) show an example of the 2009 flood:

Figure 4: **A** Flood 2009 in Middle Draa Valley; **B** Flood 2009 (Karmaoui A)

### 2.1.3 Guelmim sub-catchment

In the Guelmim sub-catchment, the drainage system is composed by Seyad Wadi, Oum El Achar Wadi, Ourg Wadi and Assaka Wadi. It is therefore subject to the risk of flooding due to overflowing of these Wadis. The main tributaries of the Assaka Wadi (Seyad and Oum Laachar) have been planning for the spreading of flood waters. Thus, seven thresholds derivation (small dam) concrete, masonry or gabions have been constructed and are used to derivate flow rates of 15 to 30 $m^3$ /s per small dam, of a total capacity of derivation of 174 $m^3$/s (Water and Environment Ministry).

On the 7[th] of January 1985, the Province of Guelmim was hit by flooding due to overflowing of the Oum Laachar Wadi, 33mm in Guelmim center and 65 mm in Bouizakarne in 53min, at a flow rate of 1000$m^3$/s(EVICC 2011b). The importance of these events is due to the socio-economical vulnerability of this area. Unfortunately, human and economic damages for these floods are not available.



### 2.1.4. Tata sub-catchment

Tata province has experienced several floods. History indicates that the floods that overflow Akka River in 1995 is one of the major events (see Table 3) that impacted the zone. The damage is as following: 13 deads, 2 wounded, 4 missing and 350 families homeless; as well as destruction of 655 homes (EVICC 2011b).

Table 3: Characteristics of the 1995 flood in Tata

### 3. Methodology

In this paper, we used and normalized different variables (indicators) for each selected spatial scale (at urban area and sub-catchment), in a numerical index that reflects the status of a region's flood vulnerability (See Table 4). The used tool or the flood vulnerability index, developed in 2009 by Balica et al. The overall FVI for each scale was determined by the calculating the index from the 36 indicators for sub-catchment scale and 34 for urban scale. These indicators have been linked with the three factors of vulnerability: susceptibility, exposure and resilience. The general FVI Eq. (1) links the values of all indicators to flood vulnerability components (social, economic, physical and environmental) and factors (susceptibility, exposure and resilience), without balancing or interpolating from a series of data.

$$FVI= \frac{Exposure * Susceptibility}{Resilience} \quad (1)$$

Table 4: Relationship between components and factors (Balica *et al.* 2009)

The calculation of each component, both in the urban and sub-catchment scale are based on the following equations (Balica & Wright 2010):

**Urban scale:**

- Social component:

$$FVI\,S = \left[ \frac{PD,\ P\ FA,\ CH,\ PG,\ HDI,\ CM}{PE,\ A/P,\ CPR,\ S,\ WS,\ E\ R,\ ES} \right] \quad (2)$$

- Economic component:

$$FVI\,Ec = \left[ \frac{IND,\ CR,\ UM,\ Ineq,\ UG,\ RT}{FI,\ AmInv,\ D\_S\ C,\ D} \right] \quad (3)$$

- Environmental component :

$$FVIEn = \left[ \frac{UG,\ Rainfall}{E\ V,LU} \right] \quad (4)$$

- Physical component :

$$FVIPh = \left[ \frac{T,\ C\ R}{E\ V/Rainfall,\ S\ C/V_{year},\ D\_L} \right] \quad (5)$$

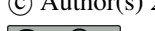



**Sub-catchment scale**

- Social component:

$$FVI\ S = \left[ \frac{P\ FA,\ R\ Pop,\ \%\ disable,\ C\ m}{P\ E,\ A/P,\ C\ PR,\ WS,\ E\ R,\ HDI} \right] \quad (6)$$

- Economic component:

$$FVI\ Ec = \left[ \frac{L,\ U\ M,\ Ineq,\ U\ A}{LEI, FI,\ AmInv,\ S\ C/V_{year},\ ECR} \right] \quad (7)$$

- Environmental component:

$$FVIEn = \left[ \frac{Rainfall,\ DA,\ UG}{L\ U,\ E\ V,\ N\ R,\ Unpop} \right] \quad (8)$$

- Physical component:

$$FVIPh = \left[ \frac{T}{E\ V/Rainfall,\ S\ C/V_{year},\ D\_L} \right] \quad (9)$$

Data for calculating the FVI (and initially setting the response levels) were collected for five oasis provinces: Guelmim, Tata, Zagora, Ouarzazate and Errachidia to provide some initial testing of the model. These data were obtained from national and regional reports.

### 3.1 Method for scoring: Standardization/normalization

A standardization method was used for adjusting indicator values in a scale from 0 to 1 (See Table 5).The standardized formula of the FVI is as follow (10):

$$FVI_S = \frac{FVI_{Scale}}{FVI_{Max}} \quad (10)$$

The flood vulnerability analysis was done using a detailed evaluation of the four components of flood vulnerability: social, economic, environmental and physical. These components were gathered and calculated to give the overall vulnerability.

Table 5: Flood vulnerability designations (Balica et al, 2012)

### 4.   Results

After data collection, the flood vulnerability index was calculated for the four flood vulnerability components and for the total FVI. The indicator values are gathered, compiled and standardized for five case studies selected of Moroccan pre-Sahara region, while the total FVI was determined by the calculating it from the 36 indicators of each sub-catchment and 34 for each urban area.

### 4.1 At urban scale of the Moroccan Pre-Sahara

At urban scale, we used the following equations: **Eq. (2)** for social component, **Eq. (3)** for economical component, **Eq. (4)** for environment and **Eq. (5)** for physical component. These equations lead to obtain the results in Fig. 5.





Figure 5: Flood vulnerability index of the five urban areas; The Social, Economic, Environmental, and Physical components and the Total FVI

The Figure 5 shows a comparison (Normalized values) between the five urban areas selected, Ouarzazate city in Upper Draa Valley (Ouarzazate province), Zagora city in Middle Draa Valley (Zagora province), Tata city in Lower Draa Valley (Tata province), Guelmim city in Guelmim province and Errachidia city in Tafilalt or (Errachidia province), for the 4 components and the total FVI.

#### 4.1.1 Social component

Fourteen indicators are used to determine the social FVI values as showed in Table 3 presented by U in social component. These indicators emphasize clearly that Zagora is very high vulnerable to floods because this city is characterized by widespread poverty, high rates of unemployment and illiteracy. During floods, planning and measures taken in this city are inadequate. As to the Errachidia city, is small vulnerable and Guelmim is very small vulnerable to floods. Regarding the Ouarzazate and Tata cities, are vulnerable to flood.

#### 4.1.2 Economic Component

The value for the FVI ec is near to zero except for the Tata city. The economic component is largely sensitive to the value of storage capacity of the area because the retention capacity improves the resilience of the cities. In fact, Tata city is very high vulnerable to floods, whereas the other cities have very small vulnerability to floods. The high vulnerability of Tata city is due to the low capacity to water storage. The other four case studies present very low values of exposure. In fact, this relative small vulnerability or resilience is due to the relative higher capacity of storage of water. Effectively, the four cities (Guelmim, Zagora, Ouarzazate and Errachidia) take advantages of the following dams (www.water.gov.ma) and small reservoirs (Ouhajou 1996).

- Mansour Eddahbi Dam (560 Mm³) near the Ouarzazate city and in upstream of the Zagora city and 5 small reservoirs in Zagora province: Agdez (3.14 m³/s), Tansikht (6.77 m³/s), Ifly (11 m³/s), Azghar (3.3-11 m³/s) and Bounou (4 m³/s), where Agdez, Tnasikht and Ifly are in upstream side of the Zagora city.
- Plus Mansour Eddahbi Ouarzazate province has a second dam called Tiouine (100 Mm³).
- Hassan Edakhil (347.0 Mm³) in upstream of the Errachidia city, and
- Seven thresholds derivation (small dams) concrete, masonry or gabions have been constructed and are used to derivate flow rates of 15 to 30 m³/s per small dam, for a total capacity of derivation of 174 m³/s in Guelmim province.

#### 4.1.3 Environmental Component

According to Fig. 5, Zagora (0.8) and Ouarzazate (1) have very high vulnerability to floods, while Tata (0.08) and Guelmim (0.013) have small vulnerability to floods, however Errachidia (0.001) is very small vulnerable to floods. The low vulnerability of Tata, Guelmim and Errachidia is due to the followings: low rainfall, low land use and high evaporation/rainfall rate. Comparing these cities with Zagora and Ouarzazate, the environmental vulnerability of Zagora and Ouarzazate to floods is higher than the three other cities. Figure 5 illustrates that Ouarzazate city has a higher environmental vulnerability to floods due to a relative large rainfall amounts and the low percentage of green areas.



### 4.1.4 Physical Component

Guelmim (1) has a very high vulnerability to floods; whereas, Errachidia (0.2), Tata (0.02), Zagora (0.05) and Ouarzazate (0.01) are classified as small vulnerable to floods.

### 4.1.5 Total Flood Vulnerability Index (FVI $_{total}$)

All the components together give the total value of the flood vulnerability index for each case study (each urban area). Errachidia (0.09) is small vulnerable, while the four other cities (Zagora (0.47), Tata (0.28), Ouarzazate (0.26), and Guelmim (0.25)) are vulnerable to floods. Comparing these four later cities, the total FVI makes Zagora more vulnerable urban area then Tata, Ouarzazate and Guelmim. Flood management leads to decrease the flood impacts of the socio-economical sector in the pre-Saharan region. The traditional management widely observed in developing countries and vulnerable region take account mainly the economic loses than the environmental and social components. As a measure to the recovery after a flooding event, is the flood insurance, this provides compensation for losses caused by the flood. Particularly, in the case studies, the insurance is not included in the flood risks. The second measure is rehabilitation. The post-flood management problems can be pre-planned. In order to achieve this, objective surveys need to be carried covering human casualties and material damage. On the basis of an objective assessment of hazard, economic, social, and environmental factors, the government should impose that the future development projects will be compliant with the local flood vulnerability.

### 4.2 At Sub-catchments scale of the Moroccan Pre-Sahara

The calculating of the FVI at sub-catchment scale in Moroccan Pre-Sahara requires a total of 36 indicators, presented in Table 2. Using equations **Eq. (6)** for social component, **Eq. (7)** for economic component, **Eq. (8)** for environmental and **Eq. (9)** for physical component, the results obtained are shown in Fig. 6.

Figure 6: Flood vulnerability index of the five sub-catchments areas for the Social, Economic, Environmental, and Physical components and for the Total FVI

Figure 6 shows a comparison (Normalized values) between the five sub-catchments areas selected, Upper Draa Valley, Middle Draa Valley (Zagora province), Tata sub-catchment (Lower Draa Valley or Tata province), Guelmim sub-catchment in Guelmim province and Errachidia sub-catchments city in Tafilalt or (Errachidia Province), for the 4 components (social, economic, environmental, and physical) and the total FVI.

### 4.2.1 Social component

Comparing the five sub-catchments, Guelmim (1) is socially the most vulnerable to floods (very high vulnerability).This high vulnerability is due to the high number of rural population, most disabled people, and the non-functional warning system than the others sub-catchments. Regarding the other sub-catchment, the MDV (0.37) and Errachidia (0.17) are vulnerable to floods, but Tata (0.04) and UDV (0.03) have small vulnerability.

### 4.2.2 Economic component

Guelmim (1) and Errachidia (1) have very high vulnerability; whereas Tata (0.16) has small vulnerability. Regarding UDV (a value close to 0) and MDV (0.001) have very small vulnerability, with small difference between



both. The storage capacity has a large influence on the economic FVI component. Its economic FVI reflects that it is not economically vulnerable to floods. Industrial facilities are very small and not vulnerable.

### 4.2.3 Environmental component

UDV (1) and Errachidia (1) have similar values, showing that they are very high vulnerable to flood due to the effect of large terrain varying between 4000 and 1000 (m.asl). MDV (0.64) has high vulnerability, whereas Guelmim (0.03) and Tata (0.018) have small vulnerability to flood, these values can be explained by the low anthropogenic influence. These values can help the flood vulnerability analysis to define strategies for the reduction of the environmental FVI for the areas having the very high vulnerability.

### 4.2.4 Physical component

Four indicators are used to determine the values of physical component of the five case studies. The UDV (1) sub-catchment is physically the highest vulnerable to flood, while MDV (0.5) being vulnerable, Guelmim (0.13) having small vulnerability, while Tata (0.065) and Errachidia (0.039) being very small vulnerable to flood.

### 4.2.5 Total Flood Vulnerability Index

The components (social, economic, environmental, and physical) together give the total value of the flood vulnerability index of each sub-catchment. The Errachidia (0.55), Guelmim (0.54), and UDV (0.5), have very similar values. These values lead to classify the three sub-catchments as high vulnerable to floods. Regarding the two other sub-catchments, can be seen that MDV (0.37) is vulnerable, and Tata (0.073) has small vulnerability to flood. Comparing the whole components, the Errachidia is the most vulnerable, with the exception of the physical component. Developing plans to reduce these components may reduce the total FVI of the Errachidia.

Comparing the flood vulnerability of the urban areas with the sub-catchments, we see that at urban scale in social component the government should reduce vulnerability in Zagora firstly and Errachidia secondly, and reduce economic vulnerability in Tata, environmental vulnerability in Ouarzazate and Zagora and physical vulnerability in Guelmim and then in Errachidia. However, for sub-catchment the government should firstly reduce vulnerability in Guelmimand secondly in the Middle Draa Valley, and reduce economic vulnerability in Guelmim and Errachidia sub-catchments, environmental vulnerability in the Upper Draa Valley (UDV) and Errachidia sub-catchments and afterwards in Middle Draa Valley (MDV) and physical vulnerability in UDV and then in MDV.

### 5. Discussion

The challenges posed by climate change (CC) increase the importance of adaptation in Moroccan pre-Saharan region. Reduced vulnerability and adaptation to CC cannot be achieved by one sector alone, but all sectors that depend directly or indirectly on services provided from this environment. The impacts of these practices include the loss of land and other natural resources (loss of biodiversity and reduced agricultural). Together, these effects cause a deterioration of living conditions and poverty especially in the rural population. Today, CC aggravates these problems. This paper also provides basic management of flood risk and informs decision makers in development and urban planning. At urban scale, the social vulnerability is very high, and government should reduce vulnerability in Zagora and Errachidia, but also reduce economic vulnerability in Tata, environmental





vulnerability in Ouarzazate and Zagora and physical vulnerability in Guelmim and in Errachidia. For the sub-catchment scale, the government should create plans to reduce vulnerability in Guelmim and Middle Draa Valley, and mitigate economic vulnerability in Guelmim and Errachidia sub-catchments, environmental vulnerability Upper Draa Valley (UDV) and Errachidia sub-catchments and only then in Middle Draa Valley (MDV) and physical vulnerability in UDV and in MDV.

To increase the socio-economic level of the poorest people, the government must invest in public transport, education (schools), appropriate housing (economical). Reducing social inequalities in flood vulnerability is the right thing to do. Reducing vulnerability also fight existing socio-economic problems. Because, reducing vulnerability is an interdisciplinary problem. It requires that physical, social and economic scientists and engineers work together to take the lead on flood vulnerability issues.

The methodology used in this paper, is based on several indicators for different factors and two geographical scales, focusing on fluvial and urban floods. Various indicators were taken into account to assess flood vulnerability.

The FVI allow to give solutions by identifying the most and the less vulnerable geographical scale in different sectors (economy, social, physical and environment), and to also bring out an easy to use tool which can be applied and used by the non-scientific community. The results of the FVI study allow that increased knowledge of these sectors can help to assess and manage probable floods. In this way, the FVI helps to identify the exact areas of potential vulnerability for the particularity or elements at risk disregarding of the intensity of the flood, which may occur. The FVI approach attempted to accomplish to take in social science knowledge to define the index indicators individually and thus calculate their vulnerability. As a reminder, the limitation of existing work shows that most collected data is descriptive. That is because most data is gathered and stored in a different ways and formats as this in turn can make comparisons difficult. Therefore, data computation and preparation for such assessments helps to derive higher accuracy. The FVI assessment demonstrated that the FVI tool can be applied at arid zone (Moroccan pre-Sahara), and can generate a range of information to help implement infrastructure projects and to identify areas of risk. The FVI study provides insights the natural and social susceptibility to flood for the four dimensions of the social, economic, environmental and physical for the urban scale. The results of the FVI study could be used for planning of new or better protected settlement in the area.

Economic development is often associated with pressures on ecosystems and ecosystem services by the mean of the overuse of forest woods, the urban development, water shortage etc. To fight against the effects of floods, two types of measures must be taken (structural and non-structural measures). Several structural measures can be taken such as dams and dikes. Planting trees in the upstream area of each sub-catchment can be seen as a method to protect and combat soil erosion. However, the non-structural measures are the actions like; response, preparedness, warning systems, rehabilitation planning, and flood fighting etc. The oases zones of Morocco are located near the Riverto facilitate the use of surface water. This location near the watercourse beds causes the exposure of these areas to flood risk. In fact, the oasis is both with high agricultural values, ecological, landscape and cultural and territories weakened. Obviously as in other countries, reducing hazard, while minimizing impacts on the natural environment and the socio-economic sector, through the construction of dams and facilities can maintain flood



prone areas. The current policy, construction of big and small dams, against flooding in the oasis basins allows the reduction of the hazard of flooding.

## 6. Conclusion

The methodology used in this paper, is based on several indicators for different factors and two geographical scales, focusing on fluvial and urban floods. Various indicators were taken into account to assess flood vulnerability. The Flood vulnerability Index was used on 5 case studies at two scales (sub-catchment scale and area scale). Lot of data was needed to estimate the flood vulnerability indicators for each of the two areas. An accurate assessment of flood vulnerability is difficult, due to the lack of official necessary data.

**At urban scale,** the Errachidia (0.09) is small vulnerable, while the four other cities (Zagora (0.47), Tata (0.28), Ouarzazate (0.26), and Guelmim (0.25)) are vulnerable to floods.

Regarding the **sub-catchment scale** the Errachidia (0.55), Guelmim (0.54), and UDV (0.5), have very similar values. These values lead to classify the three sub-catchments as high vulnerable to floods. Regarding the two other sub-catchments, can be seen that MDV (0.37) is vulnerable, and Tata (0.073) has small vulnerability to flood. Comparing the whole components, the Errachidia is the most vulnerable, with the exception of the physical component.

Concerning the applicability of Flood Vulnerability Index methodology can be summarised as follows:

- Vulnerability can be reflected by exposure, susceptibility and resilience factors;
- The sub-catchment and urban systems can be damaged regarding four different components estimated in the result section.
- The FVI is adaptable to different uses in the Moroccan pre-Saharan region.
- This tool allow to identify the risks and the management methods to assess flood vulnerability;
- Find out the priority components and sectors of flood vulnerability in order to take urgent measures
- The FVI is applicable in sub-catchment and urban area scales in pre-Saharan region.
- Finally, the proposed methodology to calculate a FVI provides an approach to quantify how much floods are affecting, or can affect a sub-catchment or an urban area in pre-Saharan regions.

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

EVICC, 2011b. Evaluation de la vulnérabilité et des impacts du changement climatique dans les oasis du Maroc et structuration de stratégies territoriales d'adaptation. Mission 1.1 : Bilan-Diagnostic des vulnérabilités climatiques et des capacités d'adaptation en situation actuelle. Rapport complémentaire - Octobre 2011.

EVICC, 2011c : Evaluation de la vulnérabilité et des impacts du changement climatique dans les oasis du Maroc et structuration de stratégies territoriales d'adaptation. Mission 1.1 : Bilan-Diagnostic des vulnérabilités climatiques et des capacités d'adaptation en situation actuelle Bassin oasien du Ziz- Tafilalet -Mai 2011

EVICC, 2011d. Evaluation de la vulnérabilité et des impacts du changement climatique dans les oasis du Maroc et structuration de stratégies territoriales d'adaptation. Mission 1.1 : Bilan-Diagnostic des vulnérabilités climatiques et des capacités d'adaptation en situation actuelle Bassin oasien de Dadès- Draa -Mai 2011

EVICC, 2011e. Evaluation de la vulnerabilite et des impacts du changement climatique dans les oasis du Maroc et structuration de strategies territoriales d'adaptation. Mission 1.1 : Bilan-Diagnostic des vulnerabilites climatiques et des capacites d'adaptation en situation actuelle. Bassin oasien de Guelmim - Tata -Mai 2011.

Fuchs S, Kuhlicke C, Meyer V, 2011. Editorial for the special issue: vulnerability to natural hazards—the challenge of integration. Nat Hazards 58: 609- 619.

HCP, 2010. LE MAROC DES REGIONS *2010.* ROYAUME DU MAROC. HAUT- COMMISSARIAT AU PLAN

Heidecke C & Roth, "Drought Effects on Livestock Husbandry" in IMPETUS Atlas Morocco. Research Results 2000–2007. 3rd Edition, edited by Schulz, Oliver and Judex, Michael, Department of Geography, University of Bonn, Germany, (2008).

IPCC, 2001. Intergovernmental Panel on Climate Change. Climate change 2001: Impacts, Adaptation and Vulnerability. Cambridge University Press, Cambridge, UK.

IPCC, 2012., Managing the risks of extreme events and disasters to advance climate change adaptation. Special report of the Intergovernmental Panel on Climate Change.*Intergovernmental panel on climate change* : 978-1-107-02506-6.

Kabiri, 2012. Desertification trends and local action in the oases of Tafilalt, South-East Morocco. Encounters Across the Atlas : Fieldtrip in Morocco 2011. http://hdl.handle.net/10138/37939). ISSN 0786-2172.

Klose A, 2009. Soil characteristics and soil erosion by water in a semi-arid catchment (WadiDrâa, South Morocco) under the pressure of global change. AngefertigtmitGenehmigung der Mathematisch-NaturwissenschaftlichenFakultät der Rheinischen Friedrich-Wilhelms-Universität Bonn.

MEA, 2005. Ecosystems and Human Well-being: Current State and Trends. *Dryland Systems*, David Niemeijer et al; 2005.

Minoia, P., & Kaakinen, I. (2012). Encounters Across the Atlas: Fieldtrip in Morocco 2011.



Molino, S., 2012. Report of the 2012 North East Victoria Flood Review, OFFICE of the Emergency Services Commissioner , Trim Id: Cd/12/448164*, October 2012.

Mortimore, M. with contributions from S. Anderson, L. Cotula, J. Davies, K. Faccer, C. Hesse, J. Morton, W. Nyangena, J. Skinner, and C. Wolfangel., 2009. Dryland Opportunities: A new paradigm for people, ecosystems and development, IUCN, Gland, Switzerland; IIED, London, UK and UNDP/DDC, Nairobi, Kenya. x + 86p.

Ouhajou L, 1996. Espace Hydraulique et Société au Maroc. Cas des Systèmes d'Irrigation dans laVallée du Dra (Doctoral thesis). Agadir: Université Ibn Zohr.

PACC, 2011- Projet d'adaptation au changement climatique au Maroc : vers des oasis résilientes - mission 1 : mise en place d'un système d'alerte et de vigilance et d'alerte contre les risques climatiques dans les régions des oasis du sud du Maroc. Composante Guelmim – Tata. Mai 2011

PACC, 2012a. Projetd'adaptation au changement climatique au Maroc: vers des oasis résilientes. Cap sur la résilience. Vers l'adaptation des territoires oasiens. Note de communication. MAI 2012.

PACC, 2012b. Projet d'adaptation au changement climatique au Maroc : Vers des oasis résilientes - mission 1 : Mise en place d'un système d'alerte et de vigilance et d'alerte contre les risques climatiques dans les régions des oasis du sud du Maroc.

RBOSM, 2008. Plan cadre de gestion de la Réserve de Biosphère des Oasis du Sud Marocain (RBOSM) ; 2008. Version provisoire.

PFRRS, 2012. (Post-flood relief and recovery survey). Cambodia: Post-flood relief and recovery survey, May 2012 (http://www.unicef.org/cambodia/)

RIAD, 2003. Typologie et analyse hydrologique des eaux superficielles 0 partir de quelques bassins versants représentatifs du Maroc. Thèse en cotutelle université des sciences et technologies de Lille & université IbnouZohr  d'Agadir.

Saidi M. Elmehdiet al, 2010., The Ourika floods (High Atlas, Morocco), Extreme events in semi-arid mountain context. *ComunicaçõesGeológicas*, 2010, t. 97, pp. 113-128.

Vatsa K & Krimgold  F, 2000 Financing disaster mitigation for the poor. In A. Kreimer, and M. Arnold (eds), Managind Disaster Risk in Emerging Economies. World Bank, Washington.


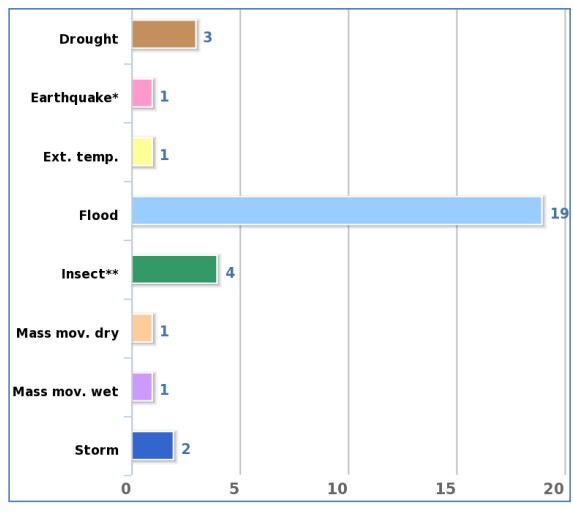

Figure 1: Natural Disaster Occurrence. Source: UNISDR (UN Office for Disaster Risk Reduction)

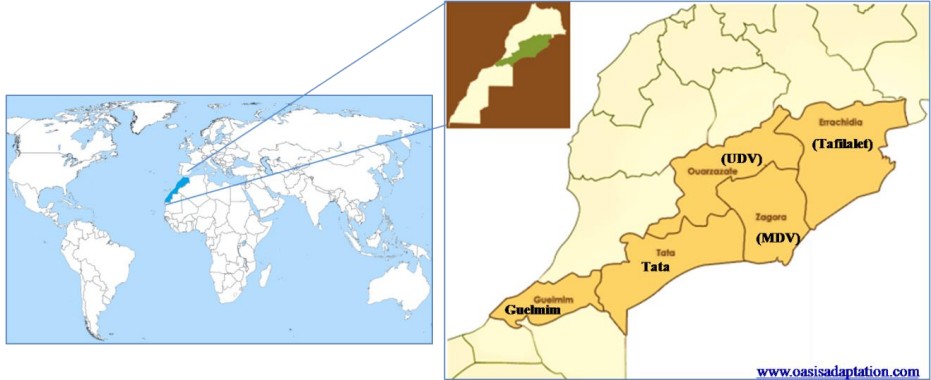

Figure 2: Moroccan pre-Sahara: Oasean zone, including the basins of Guelmim, Tata, Zagora (MDV), Ouarzazate (UDV) and Errachidia (Tafilalet)

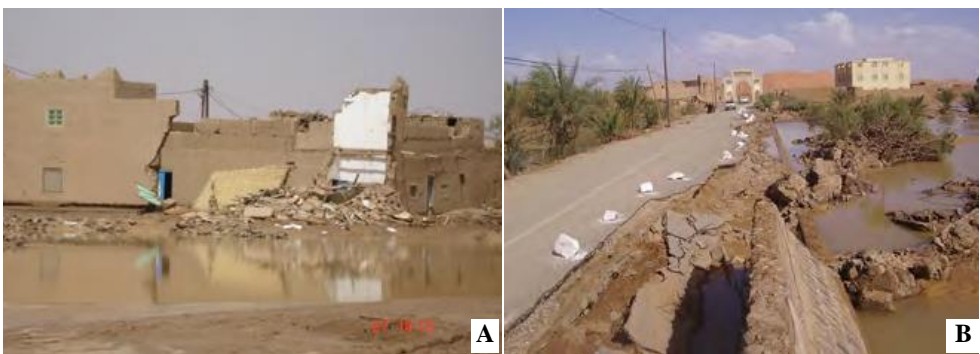

Figure 3: A: Collapsed houses in Merzouga.  B: State of the road after a flood, at the entrance of Merzouga

(Source: Kabiri, 2012).





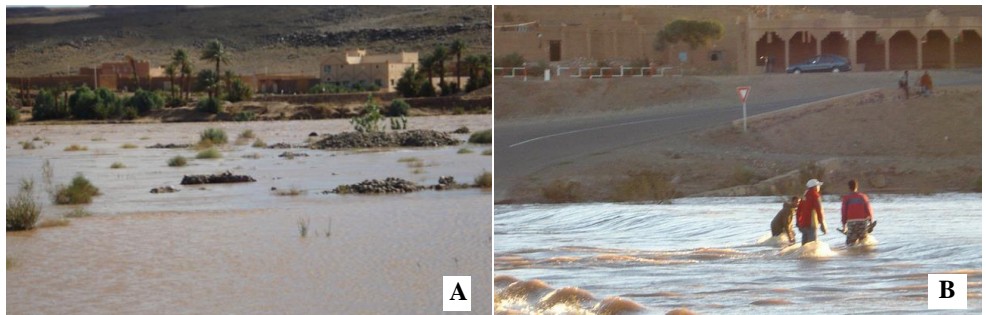

Figure 4: **A** Flood 2009 in Middle Draa Valley Figure; **B** Flood 2009 (Karmaoui A)

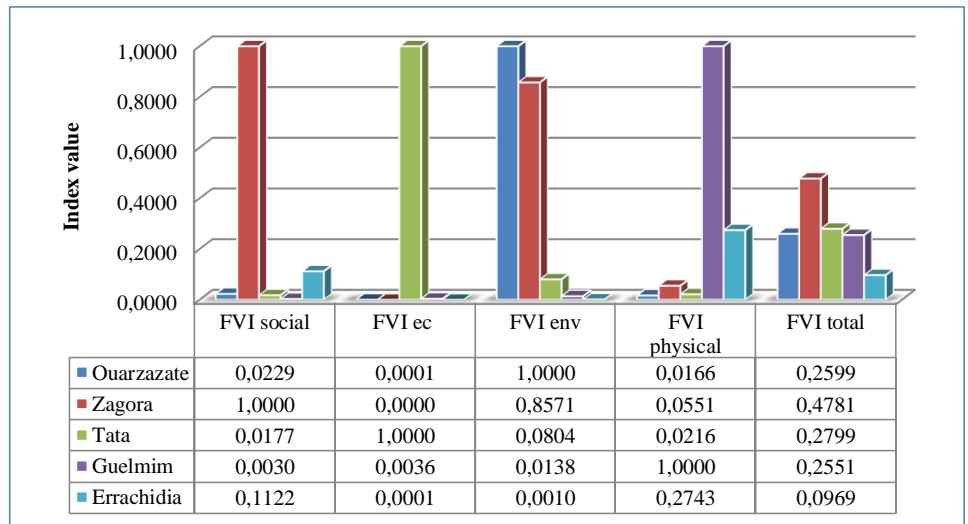

| | FVI social | FVI ec | FVI env | FVI physical | FVI total |
|---|---|---|---|---|---|
| ■ Ouarzazate | 0,0229 | 0,0001 | 1,0000 | 0,0166 | 0,2599 |
| ■ Zagora | 1,0000 | 0,0000 | 0,8571 | 0,0551 | 0,4781 |
| ■ Tata | 0,0177 | 1,0000 | 0,0804 | 0,0216 | 0,2799 |
| ■ Guelmim | 0,0030 | 0,0036 | 0,0138 | 1,0000 | 0,2551 |
| ■ Errachidia | 0,1122 | 0,0001 | 0,0010 | 0,2743 | 0,0969 |

Figure 5: Flood vulnerability index of the five urban areas; The Social, Economic, Environmental, and Physical components and the Total FVI





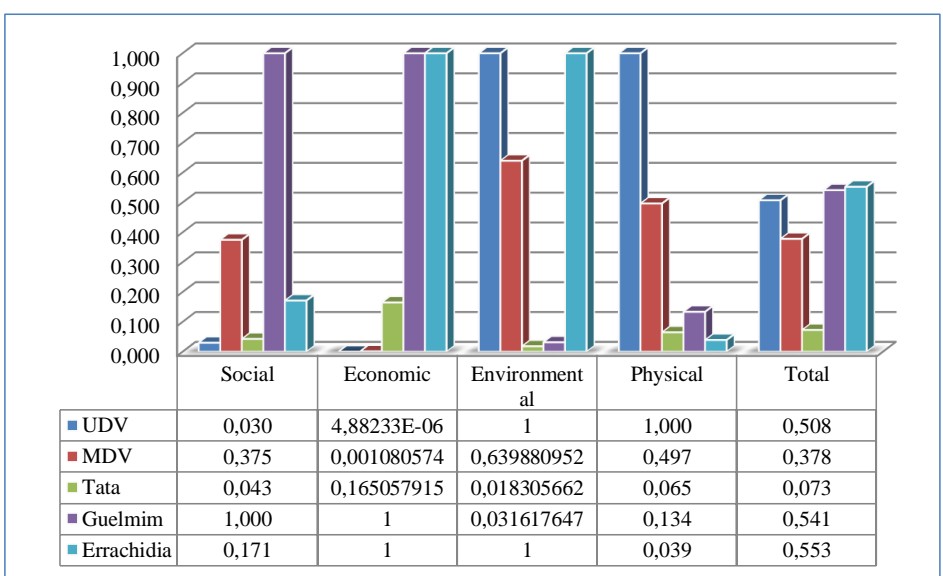

| | Social | Economic | Environmental | Physical | Total |
|---|---|---|---|---|---|
| ■ UDV | 0,030 | 4,88233E-06 | 1 | 1,000 | 0,508 |
| ■ MDV | 0,375 | 0,001080574 | 0,639880952 | 0,497 | 0,378 |
| ■ Tata | 0,043 | 0,165057915 | 0,018305662 | 0,065 | 0,073 |
| ■ Guelmim | 1,000 | 1 | 0,031617647 | 0,134 | 0,541 |
| ■ Errachidia | 0,171 | 1 | 1 | 0,039 | 0,553 |

Figure 6: Flood vulnerability index of the five sub-catchments areas for the Social, Economic, Environmental, and Physical components and for the Total FVI

Table 1: Total number of people affected since 1963 due to flood in Morocco. Source: "EM-DAT: The OFDA/CRED International Disaster Database. www.em-dat.net"

| Date | No Total Affected |
|---|---|
| 22/01/1970 | 266444 |
| 25/11/2010 | 75003 |
| 21/01/1996 | 60000 |
| 11/1965 | 47813 |
| 1977 | 38000 |
| 12/1963 | 35010 |
| 17/08/1995 | 35000 |

Table 2: Characteristics of the 1995 flood in Tata

| Rainfall level | Duration of rainfall | Debit (m³/s) | Water level major bed |
|---|---|---|---|
| 106mm | 180min | n.d | 1000mm |

Table 3: Human and economic losses from disasters occurred between 1980- 2010, in Morocco

| Killed People | | | Economic Damages | | |
|---|---|---|---|---|---|
| Disaster | Date | Killed | Disaster | Date | Cost (US$ X 1,000) |
| Flood | 1995 | 730 | Drought | 1999 | 900,000 |
| Earthquake | 2004 | 628 | Earthquake | 2004 | 400,000 |
| Flood | 2002 | 80 | Flood | 2002 | 200,000 |
| Flood | 1997 | 60 | Flood | 1996 | 55,000 |
| Flood | 1995 | 43 | Flood | 1995 | 9,000 |
| Flood | 2003 | 35 | Flood | 2001 | 2,200 |
| Flood | 2010 | 32 | Extreme temp. | 2000 | 809 |
| Mass mov. dry | 1988 | 31 | Storm | 2005 | 50 |
| Flood | 2008 | 30 | | | |
| Flood | 1996 | 25 | | | |



Table 4: Relationship between components and factors (Balica *et al.* 2009)

Overall indicators Relationship between components and factors

| Flood Vulnerability | Exposure | Geographic scale | Susceptibility | Geographic scale | Resilience | Geographic scale |
|---|---|---|---|---|---|---|
| Social Components | Population density | R.S.U | Past experience | R.S.U | Warning system | R.S.U |
| | Population in flood area | R.S.U | Education (literacy rate) | R.S.U | Evacuation routes | R.S.U |
| | Closeness to inundation area | R.S.U | Preparedness | R.S.U | Institutional capacity | R.S.U |
| | Population close to coast line | R.S.U | Awareness | R.S.U | Emergency service | R.S.U |
| | Population under poverty | R.S.U | Trust in institutions | R.S.U | shelters | R.S.U |
| | % of urban area | R.S.U | Communication penetration rate | R.S.U | | |
| | Rural population | R.S | hospitals | R.S.U | | |
| | Cadastre survey | S.U | Population with access to sanitation | R.S.U | | |
| | Cultural heritage | S.U | Rural population who access to WS | R.S | | |
| | % of young & older | S.U | Quality of water supply | S.U | | |
| | Slums | U | Quality of energy supply | S.U | | |
| | | | Population growth | S.U | | |
| | | | Human health | S.U | | |
| | | | Urban planning | U | | |
| Economic components | Land use | R.S.U | Unemployment | R.S.U | Investment in counter measures | R.S.U |
| | Proximity to river | R.S.U | Income | R.S.U | Infrastructure management | R.S.U |
| | Closeness to inundation areas | R.S.U | Inequality | R.S.U | Dams & storage capacity | R.S.U |
| | % of urbanized area | R.S | Quality of infrastructure | R.S.U | Flood insurance | R.S.U |
| | Cadastre survey | S.U | Years of sustaining health life | R.S.U | Recovery time | R.S.U |
| | | | Urban growth | S.U | Past experience | S.U |
| | | | Child mortality | S.U | Dikes/levees | S.U |
| | | | Regional GDP/Capita | S | | |
| | | | Urban planning | S | | |
| Environmental components | Ground WL | R.S.U | Natural reservations | R.S.U | Recovery time to floods | R.S.U |
| | Land use | R.S.U | Years of sustaining health life | R.S.U | Environmental concern | R.S.U |
| | Over used area | R.S.U | Quality of infrastructure | R.S.U | | |
| | Degraded area | R.S.U | Human health | S.U | | |
| | Unpopulated land area | R.S | Urban growth | S.U | | |
| | Types of vegetation | R.S | Child mortality | S.U | | |
| | % of urbanize area | R.S | | | | |
| | Forest change rate | R | | | | |


| Physical components | | R.S.U / Buildings codes | U | Dams & storage capacity |
|---|---|---|---|---|
| | Topography (slope) | R.S.U | | Dams & storage capacity — R.S.U |
| | Geography | R.S.U | | Roads — R.S.U |
| | Geology | R.S.U | | Dikes /levees — S.U |
| | Heavy rainfall | R.S.U | | |
| | Flood duration | R.S.U | | |
| | Return periods | R.S.U | | |
| | Proximity to river | R.S.U | | |
| | Soil moisture | R.S.U | | |
| | Evaporation rate | R.S.U | | |
| | Temperature (yearly average) | R.S.U | | |
| | River discharge | R.S.U | | |
| | Frequency of occurrence | R.S.U | | |
| | Flow velocity | S.U | | |
| | Storm surge | S.U | | |
| | Tidal | S.U | | |
| | Flood water depth | S.U | | |
| | Sedimentation load | S.U | | |
| | Coast line | S.U | | |
| | Coastal bathymetry | S.U | | |

Where: **R**, represents River Basin Scale; **S**, represents Sub-catchment Scale; and **U**, represents Urban Scale



Table 5: Flood vulnerability designations (Balica et al, 2012)

| Index value | Designations |
|---|---|
| <0.01 | Very small vulnerability to floods |
| 0.01 to 0.25 | Small vulnerability to floods |
| 0.25 to 0.5 | Vulnerable to floods |
| 0.5 to 0.75 | High vulnerability to floods |
| 0.75 to 1 | Very high vulnerability to floods |





**Appendix 1: Questions applied to the interview**
(Adapted from Molino, S., 2012 and PFRRS, 2012)

**SAMPLE OF STAKEHOLDER INTERVIEW QUESTIONS**

1. What was your organisation's role in floods?

2. What was your role in the incident?

3. Was a seamless and integrated approach provided to the community by emergency management organisations during the incident? Yes/no (reasons)

4. What could be improved to deliver a more seamless and integrated approach?

5. Did you experience or do you know of any interoperable issues during the incident? Yes/No if so, what were they? (list issues)

6. Prior to the flood, did the community, individuals, businesses and emergency management organizations understand their roles in the event of a flood emergency? Yes/No (reasons)

7. What plans were in place to ensure the incident was well managed, coordinated and communicated?

8. Was the local plan activated during the incident? Yes/No (reasons)

9. How could the management of the incident be improved? (list ways)

10. How effectively were vulnerable people supported in the incident?

11. Were people well informed about how to access support and essential services such as shelter, food, water and medical care? Yes/No (reasons)

12. Partners interested in helping the poorest households and reducing the impact of future natural disasters

13. Additional financial support, in the form of targeted social safety net activities, is needed by the poorest and most vulnerable households to protect against the deterioration of the health and nutritional status of their families, particularly children.

14. Was this provided in a timely way? Yes/No (reasons)

15. Did people continue to receive essential and critical services during the incident? Yes/No (reasons)

16. Were the social, economic and environmental impacts rapidly assessed to develop the recovery plan? Yes/No

17) Preparedness and recovery efforts will best be directed towards hygiene education, as well as strategic prepositioning and continued distribution of water treatment materials in high-risk and flood affected areas.





**Appendix 2: Reference Data availability**
**Table A2.1. Urban area**

| | Name | Definition | Units | References |
|---|---|---|---|---|
| 1 | Population density | There is an important exposure to a given hazard if population is concentrated | people /km² | EVICC, 2011a ; 2011c, 2011d; and 2011e |
| 2 | Population in flood prone area | Number of people living in flood prone area | people | EVICC, 2011a ; EVICC, 2011e |
| 3 | Cultural Heritage | Number of historical buildings, museums, etc., in danger when flood occurs, if none take 1 | | Survey Monographic reports (2008, 2011) |
| 4 | Population growth | % of the population growth in urban areas in the last 10 years | % | Monographic reports (2008, 2011) provincial and regional www.hcp.com |
| 5 | Disabled People | % of population with any kind of disabilities, also people less 15 and more than 65 | % | www.hcp.com |
| 6 | Human Development Index | * HDI = ¹/₃(LEI) + ¹/₃(EI) + ¹/₃(GI) | | www.hcp.com |
| 7 | Child Mortality | Number of children less than 1 year old, died per 1000 births | | www.hcp.com and DHS, 2004 |
| 8 | Past Experience | # of people affected in last 10 years because floods; | people | Survey and interviews |
| 9 | Awareness &Preparedness | Range between 1-10 (help) | | Survey and interviews |
| 10 | Communication Penetration Rate | % of households with sources of information | % | www.hcp.com |
| 11 | Shelters/Hospitals | Number of shelters per km², including hospitals | #/km² | Survey and interviews |
| 12 | Warning system | If No WS than the value is 1, if yes WS than the value is 10 | | EVICC, 2011b and c |
| 13 | Emergency Service | Number of people working in this service | # | Survey |
| 14 | Evacuation roads | % of asphalted roads | % | Monographie 2011 |
| 15 | Industries | # of industries or any types of economic activities in urban area | # | Monographie 2008.. |
| 16 | Contact with River | Distance of city along the river | km | Google earth |
| 17 | Unemployment | $UM = [\,^{\#of\_people\_unempl}/_{Total\_Pop\_AptToWork}\,] * 100$ | % | www.hcp.com |
| 18 | Inequality | Gini Coefficient for wealth inequality, between 0 and 1 | | 2004 GINI:http://www.memoireonline.com/12/06/305/realisation-objectifs-du-millenaire-developpement-maroc-optimisation-spatiale.html |
| 19 | Flood Insurance | The number flood insurances per 100 inhabitants, if 0 than take 1 | | survey |


| | Name | definition | Units | References |
|---|---|---|---|---|
| 20 | Amount of Investment | Ratio of investment over the total GDP | | Calculated from regional GDP HCP, 2010 |
| 21 | Dikes_Levees | Km of dikes/levees | km | Earth google and interviews |
| 22 | Dams_Storagecapacity | Storage capacity in $m^3$ of dams, polders, etc., upsteam of the city | $m^3$ | EVICC, 2011b, 2011cand 2011d www.water.gov.ma |
| 23 | Recovery time | Amount of time needed by the city to recover to a functional operation after flood events | days | Survey |
| 24 | Rainfall | The average rainfall/year | m/year | www.water.gov.ma EVICC, 2011b, 2011cand 2011d |
| 25 | Green Area | Area destined for green areas inside the urban area | % | Google earth |
| 26 | UrbanGrowth | % of increase in urban area in last 10 years; Fast urban growth may result in poor quality housing and thus make people more vulnerable | % | Historic of googleearth |
| 27 | Evaporation rate | Yearly decrease rate in groundwater level | m/year | EVICC, 2011b, 2011cand 2011d www.water.gov.ma |
| 28 | Topography | Average slope of the city | | http://www.toutcalculer.com/batiment/calculer-une-pente.php |
| 29 | RiverDischarge | Maximum discharge in record of the last 10 years, $m^3/s$ | $m^3/s$ | www.water.gov.ma |
| 30 | Evaporation rate/Rainfall | Yearly Evaporation over yearly rainfall | | Calculated from 24 and 27 indicators |
| 31 | Dams_Storagecapacity | The total volume of water, which can be stored by dams, polders, etc. (amount of storage capacity) | $m^3$ | www.water.gov.ma EVICC, 2011b, 2011cand 2011d |
| 32 | Drainage system | Km of canalization in the city | km | Survey and interviews |
| 33 | Average RiverDischarge | Average Riverdischarge at the mouth | $m^3/s$ | www.water.gov.ma |
| 34 | Storage capacity over yearly discharge | Amount of storage capacity over the yearly average runoff volume | | EVICC, 2011b, 2011cand 2011d www.water.gov.ma |

**Table A2.2. Sub-catchment area**

| | Name | definition | Units | References |
|---|---|---|---|---|
| 1 | Population density | There is an important exposure to a given hazard if population is concentrated | people/km² | EVICC, 2011a ; 2011c; & 2011d; and 2011e |
| 2 | Population in flood prone area | Number of people living in flood prone area | people | EVICC, 2011a ; EVICC, 2011e |
| 3 | Urbanized Area | % of total area which is urbanized | % | |



| # | Name | Description | Unit | Source |
|---|---|---|---|---|
| 4 | Rural population | % of population living outside of urbanized area | % | EVICC, 2011a ; EVICC, 2011e |
| 5 | Disabled People | % of population with any kind of disabilities, also people less 15 and more than 65 | % | www.hcp.com |
| 6 | HumanDevelopment Index | $HDI = 1/3(LEI) + 1/3(EI) + 1/3(GI)$ | | www.hcp.com |
| 7 | Child Mortality | Number of children less than 1 year old, died per 1000 births | | www.hcp.com EVICC, 2011c, 2011d, 2011e, |
| 8 | PastExperience | # of people affected in last 10 years because floods; | people | Survey and interviews |
| 9 | Awareness&Preparedness | Range between 1-10 (help) | | Survey and interviews |
| 10 | Communication Penetration Rate | % of households with sources of information | % | www.hcp.com |
| 11 | Warning system | If No WS than the value is 1, if yes WS than the value is 10 | | EVICC, 2011a ; EVICC, 2011e |
| 12 | Evacuation Roads | % of asphalted roads | % | |
| 13 | Proximity to river | average proximity of populated areas to flood prone areas | km | Google earth, survey and interviews |
| 14 | Land Use | % area used for industry, agriculture, any types of economic activities | % | Land use |
| 15 | Unemployment | $UM = [ \#of\_people\_unempl / Total\_Pop\_AptToWork ] * 100$ | % | www.hcp.com |
| 16 | Inequality | Gini Coefficient for wealth inequality, between 0 and 1 | | 2004 GINI: http://www.memoireonline.com/12/06/305/realisation-objectifs-du-millenaire-developpement-maroc-optimisation-spatiale.html |
| 17 | Life expectancy Index | $LEI = (LE - 25) / (85 - 25)$ | | |
| 18 | Flood Insurance | The number flood insurances per 100 inhabitants, if 0 than take 1 | | Survey and interviews |
| 19 | Amount of Investment | Ratio of investment over the total GDP | | Calculated from regional GDP HCP, 2010 |
| 20 | Dikes_Levees | Km of dikes/levees over total length of river | % | Google earth and survey |


| | | | | |
|---|---|---|---|---|
| 21 | Dams_Storage capacity | Amount of storage capacity over area of sub-catchment | m | www.water.gov.ma |
| 22 | Economic Recovery | How affected is the economy of a region at a large time scale, because of floods | | Survey and interviews |
| 23 | Rainfall | The average rainfall/year of a whole RB = mm / (1000 * year) = m / year | m/year | EVICC, 2011b, 2011cand 2011dwww.water.gov.ma |
| 24 | Degrated Area | % of degraded area | % | Land use |
| 25 | UrbanGrowth | % of increase in urban area in last 10 years; Fast urban growth may result in poor quality housing and thus make people more vulnerable | % | Historic googleearth |
| 26 | Forested Area | % of forested area | % | Monographie, 2008 and 2011 EVICC, 2011b, 2011cand 2011d |
| 27 | Evaporation rate | Yearly evaporation rate | m/year | www.water.gov.ma EVICC, , 2011b, 2011c and 2011d |
| 28 | Natural Reservation | % of natural reservation over total SC area | % | |
| 29 | UnpopulatedArea | % of area with density of population less than 10 pers/km$^2$ | % | |
| 30 | Topography | Average slope of sub-catchment | | MNT and http://www.toutcalculer.com/batiment/calculer-une-pente.php |
| 31 | RiverDischarge | Maximum discharge in record of the last 10 years, m$^3$/s | m$^3$/s | www.water.gov.ma |
| 32 | Frequency of occurrence | Years between floods | years | |
| 33 | Evaporation rate/Rainfall | Yearly Evaporation over yearly rainfall | | www.water.gov.ma |
| 34 | Dams_Storagecapacity | The total volume of water, which can be stored by dams, polders, etc. (amount of storage capacity) | m$^3$ | www.water.gov.ma |
| 35 | Average RiverDischarge | Average Riverdischarge at the mouth | m$^3$/s | www.water.gov.ma |
| 36 | Storage capacity over yearly discharge | Storage capacity divided by yearly volume runoff | | www.water.gov.ma |