# Peer review of "Analysis of applicability of flood vulnerability index in Pre-Saharan region, a pilot study to assess flood in Southern Morocco"

_Natural Hazards and Earth System Sciences, 2016_

## Referee Comment (RC1) · Anonymous Referee #1 · 9 May 2016

Dear authors, I have read the paper with interest. It is an informative text on the application of an existing method. The idea is to provide an assessment on a case study, in Southern Marocco, and promote the interest of use a so-called composite indicators, here the FVI - Flood Vulnerability Index. If the method is then not new, the case study could remain relevant and the subject matter would be of major interest only if this paper is substantially revised. Lastly, the paper needs major rewriting; there are multiple grammar errors and language grasp. In my view, this paper requires major revision to be published.

[specific comments] First of all, I would suggest a simpler title by removing [to assess flood], that is obvious with FVI in the first part. I had difficulties with the abstract, which

should be rewritten. It has to point out shortly the arguments that explain why is it interesting and relevant to apply FVI in this case. In general, the text is quite poorly written, with too long sentences, sometimes incomplete and needs major rewriting for sense and flow. The introduction has to be reformulated as well. Try to explain better or make more explicit the links what you are deal with. For example, you start with dry lands and droughts. Would it be more efficient just to explain that CC may increase extreme events (like drought and floods) in an area already affected by natural hazards? And then, as floods seem the most impacting hazard, you have decided to focus on it. . .. It is often the case that you suggest unclear causal relationship (date palm trees die, but why ? Precise the link with floods,. . .). In addition, you wait a bit too much before defining vulnerability. That makes imprecise the employment of this term before (sometimes used in the singular or plural forms). In general, the vocabulary is not enough precise "floods are the most dangerous natural disasters". Dangerous doesn't suit very well and, moreover, if you mean that in terms of affected people and damages, you have to refer to table 2 (and not only fig 1 and table 1). Before the fig 1, there is a sentence within any previous link. Explain why suddenly you speak about dams. The end of the introduction (the last 8 lines) needs to be clarified and the sequence of tenses (here and in the other part as well) deserves to be considered. Some repetitiveness could be removed (in fact,. . .),that easily can be fixed. In the Materials and Methods section, do you mean "benchmark" instead of indicator in the first sentence? It sounds clearer to me. At this stage, the article is not very well-structured as you find the description of the studied sites and nothing about the materials (the indicators or sources of data you will use. . .) or methods. It appears after, in the section "methodology". This has to be revised and reorganized. Table 3 is useless. Just give this information in one sentence directly in the text, as you did before, with the other examples. In the methodology part, it was not easy for me to link the equation with the three terms (exposure, susceptibility and resilience), which are not defined – and the four components. The table 4 clarifies that, but a description in the text should be given. In the description of each component by unit (urban scale or sub-catchment), be careful about you use of the vulnerability

term (it's only a part of the vulnerability which is highlighted. . .the social part, or physical and so on). Concerning the description of procedures , they could still be improved. Indeed, they are a bit unclear or some parts are missing; it would be difficult for others to reproduce the study by reading the article. You shoud precise shortly how you treat the data like the missing values, which kind of aggregation method you used, did you apply some ponderations etc. . . (you will find here some indications from the Handbook on composite indicators provided by the OECD/JRC, about how they describe the differents steps : http://composite-indicators.jrc.ec.europa.eu/?q=content/overview). Finally, a better description of procedures will help you to improve the discussion. Some study implications or limitations should be clearly presented. Concerning the adaptive measures or recommendations, it is interesting but it could be relevant to know where they come from (experts, literature, ...). The conclusion contains one of the most interesting assertion which should have lead the article :"an accurate assessement of flood vulnerability is difficult, due to the lack of official necessary data." So how did you manage to surpass this issue? How did you cover the missing data? [and please correct in the text – "data are not data is". . .and "a tool allows", sometimes you forget the "s"] is it a limitation to apply such index method? . . . Figures and tables are fine. The quality and support text are acceptable, except for the fig 2, the scale is missing and the names are not visible. I would only suggest to remove table 2 and perhaps figure 4, if there is no description attached in the text. Please just pay attention that there is a mismatch between table 2 and 3 in the text.

---

## Author Comment (AC1) · 30 May 2016

30.05.2016, Juan les Pins Dear Reviewer, We would like to thank you for your time and expertise in reading and advise us with the article. All observations and suggestions are adequately addressed. The manuscript has been rechecked to ensure spellings and punctuations are correct. The specific remarks have been marked as following: R1: First of all, I would suggest a simpler title by removing [to assess flood], that is obvious with FVI in the first part.

A: The title have been modified as suggested "Analysis of applicability of flood vulnerability index in Pre-Saharan region, a pilot study in Southern Morocco". R1: Difficulties with the abstract, which should be rewritten. It has to point out shortly the arguments

that explain why is it interesting and relevant to apply FVI in this case. In general, the text is quite poorly written, with too long sentences, sometimes incomplete and needs major rewriting for sense and flow. A: The abstract was modified into the manuscript, the relevance of why to apply the FVI was added, as well. Please, see Page 1, Lines 15-20 and 22-24. R1: The introduction has to be reformulated as well. Try to explain better or make more explicit the links what you are deal with. For example, you start with dry lands and droughts. Would it be more efficient just to explain that CC may increase extreme events (like drought and floods) in an area already affected by natural hazards? And then, as floods seem the most impacting hazard, you have decided to focus on it: : :. It is often the case that you suggest unclear causal relationship (date palm trees die, but why ? Precise the link with floods,: : :). In addition, you wait a bit too much before defining vulnerability. That makes imprecise the employment of this term before (sometimes used in the singular or plural forms). In general, the vocabulary is not enough precise "floods are the most dangerous natural disasters". Dangerous doesn't suit very well and, moreover, if you mean that in terms of affected people and damages, you have to refer to table 2 (and not only fig 1 and table 1). Before the fig 1, there is a sentence within any previous link. Explain why suddenly you speak about dams. The end of the introduction (the last 8 lines) needs to be clarified and the sequence of tenses (here and in the other part as well) deserves to be considered. A: Thank you for pointing this out. We did revised the Introduction, so changes were done as following: Page 2, Lines: 36, 38, 45-47, 49, 55-57. The unclear relationship with the date palm tree was deleted. The sentences are now shorts and complete. As suggested by the reviewer, we have reported that CC may increase extreme events (like drought and floods) in an area already affected by natural hazards. And then, as floods seem the most impacting hazard, we have decided to focus on it As commented: we have referred "floods are the most dangerous natural disasters" to table 1, table 2 and fig 1 (see the text) Page 2 Line 59 Before the fig 1, we did modified the statement about dams (Page 1, Line 63-65). Also, the latter part of the introduction (the last 8 lines) is modified to make it more scientific (Page 3, Lines 75, 83-85, 94-102).

[Figure]

R1: Some repetitiveness could be removed (in fact,: : :),that easily can be fixed. A: the "in fact" issue was fixed.

R1: In the Materials and Methods section, do you mean "benchmark" instead of indicator in the first sentence? It sounds clearer to me. At this stage, the article is not very well-structured as you find the description of the studied sites and nothing about the materials (the indicators or sources of data you will use: : :) or methods. It appears after, in the section "methodology". This has to be revised and reorganized. Table 3 is useless. Just give this information in one sentence directly in the text, as you did before, with the other examples. In the methodology part, it was not easy for me to link the equation with the three terms (exposure, susceptibility and resilience), which are not defined – and the four components. The table 4 clarifies that, but a description in the text should be given. A: The section Materials and Methods was reorganized. We have added some data on the indicators and the used sources of data. (Page 6, Lines 176-186, 189-190). Page 7, Line 199 - 201 exposed the understanding of Eq. 1 and the base of FVI methodology. Table 3 is deleted as mentioned in the review comment, this information was done in one sentence directly in the text.

R1: In the description of each component by unit (urban scale or sub-catchment), be careful about you use of the vulnerability term (it's only a part of the vulnerability which is highlighted: : :the social part, or physical and so on). Concerning the description of procedures, they could still be improved. Page 8, Lines 252-262 shows the description of procedures and aggregation method used for the total FVI.

R1: indeed, they are a bit unclear or some parts are missing; it would be difficult for others to reproduce the study by reading the article. A: This is already done by the previous changes. We believe that now the readers can easily understand the article.

R1: You shoud precise shortly how you treat the data like the missing values, which kind of aggregation method you used, did you apply some ponderations etc: : : (you will find here some indications from the Handbook on composite indicators provided by the OECD/JRC, about how they describe the different steps : http://composite-indicators.jrc.ec.europa.eu/?q=content/overview). A: Thank you very much for your suggestion. While creating the methodology this Handbook was considered. We did answer this matter in Page 8, Lines 252-262, where we pointed out why this method does not use weights. Related to the data missing values, this was also corrected into the text in Page 15, Lines 457-460.

R1: Finally, a better description of procedures will help you to improve the discussion. Some study implications or limitations should be clearly presented. A: Dear Reviewer, the limitation of this methodology in these case studies, as we indicated, it is the lack of official necessary data.

R1: Concerning the adaptive measures or recommendations, it is interesting but it could be relevant to know where they come from (experts, literature, ...). A: Two references were added Walker, G., & Burningham, K., 2011 and Khan, M. H., 2001. See text Page 14, Lines 412

R1: The conclusion contains one of the most interesting assertion which should have lead the article :"an accurate assessement of flood vulnerability is difficult, due to the lack of official necessary data." So how did you manage to surpass this issue? How did you cover the missing data? A: Please see the text in Page 15, Lines 464-469 and Appendix 2 we discuss how we deal with these data.

R1: [and please correct in the text – "data are not data is": : :and "a tool allows", sometimes you forget the "s"] is it a limitation to apply such index method? : : : A: Page 15, Lines 470-475. Overall, there is no limitation in applying FVI methodology. The FVI can show readily implicit and readily communicated results that can help decision-makers in identifying the most effective measures to be taken. Uncertainty is not removed, but is integrated into the assessment. On the other hand the complexity of FVI methodology is also a negative point, since it takes a long time and good knowledge of the area and the system behind the FVI to be able to implement it. R1:

Figures and tables are fine. The quality and support text are acceptable, except for the fig 2, the scale is missing and the names are not visible. I would only suggest to remove table 2 and perhaps figure 4, if there is no description attached in the text. Please just pay attention that there is a mismatch between table 2 and 3 in the text. A: Illustrations - Figure 2 is now clear (Page 20, Line 608), the scale was added, as well - Figure 4 (a short description was added, Page 5, Line 152).

Please also note the supplement to this comment:
http://www.nat-hazards-earth-syst-sci-discuss.net/nhess-2016-96/nhess-2016-96-AC1-supplement.pdf

**Supplement:**

**Analysis of applicability of flood vulnerability index in Pre-Saharan region, a pilot study in Southern Morocco**

A. Karmaoui[a], S.F. Balica[b] , M. Messouli[c]

[a] LHEA (URAC 33). Department of Environmental Sciences, Faculty of Sciences Semlalia, Cadi Ayyad University, Marrakesh, Morocco E-mail: Karmaoui.ahmed@gmail.com
[b] E-mail: s.f.balica@gmail.com.
[c] LHEA (URAC 33). Department of Environmental Sciences, Faculty of Sciences Semlalia, Cadi Ayyad University, Marrakesh, Morocco E-mail: messouli@gmail.com

**Abstract**

*Moroccan Pre-Saharan zone is an oasis system, which it is characterized by extreme events, like drought and flood. The flood risks will likely increase in frequency and magnitude due to global and regional climate change. Floods tend to have an important impact on isolated and poor regions such as oasis regions. This extreme event impacts are seen in the same time in social and economic sectors and accelerated by the dry land physical and environmental drivers like land use, topography, proximity to rivers. In Morocco, the use of composite indices to evaluate natural disasters is new. To reduce vulnerability cannot be accomplished by one sector alone. Therefore, it is a need to use a multidisciplinary approach to measure vulnerability, such as the Flood Vulnerability Index (FVI). This paper aims to analyze the applicability of such index in pre-Saharan region of Morocco. The FVI is a numerical index that assess the position of a region's flood vulnerability. It was determined for four components social, economic, physical, and environmental. These components can help to assist in understanding the degree of vulnerability to floods, therefore to propose strategies for improvement of the holistic system and to find out the priority components and sectors of flood vulnerability in order to take urgent measures. For this study five sub-catchments were selected: Upper Draa Valley (UDV), Middle Draa Valley (MDV), Tata sub-catchment, Guelmim sub-catchment and Tafilalt sub-catchment; and five urban areas: Ouarzazate, Zagora, Tata, Guelmim and Errachidia. A comparative analysis of the results from those areas allows us to assess the applicability of the FVI. The overall FVI for these areas was determined by the calculating and standardization of 36 indicators for each sub-catchment scale and 34 for each urban scale.*

*Keywords: flood, vulnerability, oasis, environmental impact, climate change, adaptation*

**1. Introduction**

Globally, dry land areas are estimated to be about 41 percent of the terrestrial surface, and are home to a third of humanity, and concentrate the high rate of poverty (Mortimore *et al,* 2009). Dry lands are located mainly in poor countries: 72% of this area is found within developing countries and only 28% within industrial ones (MEA, 2005). Morocco is one of these countries. Geographically is located in the North-West corner of Africa, bordered by the Mediterranean Sea and the Atlantic Ocean on the North and West, by Algeria on the East, and by Mauritania on the South. Its total land area is 710850 km$^2$ and includes different landforms, like agricultural plains, river valleys, plateaus, and mountain chains (Anon 2004). Most of these lands are arid to semi-arid from which 75% are rangelands, 13% forests and 8% are cultivated (Dahan, 2012). In the hyper-arid and arid dry lands (the desert biome), most agricultural activities are in oasis, where the irrigation is by fluvial, ground, or local water sources (MEA, 2005). This dependence of oases on water makes this area highly vulnerable to extreme events, like droughts and floods. Climate change causes acceleration in the frequency of extreme events. Human societies have developed in trying to cope by limiting impacts. In this context the IPCC (2012) has developed risk management strategies. According to the IPCC report, the impacts of changes in floods are highly dependent on how climate changes in the future (IPCC, 2012). The impact of natural disasters is correlated to the vulnerability of communities in developing countries, as previous socio-economic vulnerabilities may accelerate these disasters, making the recovery very difficult (Vatsa & Krimgold, 2000). Thus, the impact of such events increases the poverty (Carter *et al.* 2007). The climate change may increase extreme events (like drought and floods) in an area already affected by natural hazards, and then, as floods seem the most impacting hazard, we have decided to focus on it. Historically, floods have damaged properties infrastructure and thousands of populations. In Morocco, floods are the most dangerous natural disasters, as seen in Table 1, Table 2 and Figure 1. The number of affected people and lives lost due to floods exceeds any other natural disasters in the past thirty years. The data related to human and economic losses from disasters that have occurred between 1980 and 2010 in Morocco, according to UNISDR (UN Office for Disaster Risk Reduction (www.preventionweb.net), can be seen on Table 2). In order to adapt to these extreme events during this period (1980-2010), the Moroccan government built 78 dams at national scale. These dams aim mainly to control the floods (regulating service) by reducing fluctuations of the Wadis flow.

[revised manuscript text omitted]
 Figure 4) show an example of the 2009 flood that isolated several villages of Beni Zouli from the national road N 9, and then the stop of provisioning services during 15 days.

Figure 4: **A** Flood 2009 in Middle Draa Valley Figure; **B** Flood 2009 (Karmaoui A)

**Guelmim sub-catchment**

In the Guelmim sub-catchment, the drainage system is composed by Seyad Wadi, Oum El AcharWadi,Ourg Wadi and Assaka Wadi. It is therefore subject to the risk of flooding due to overflowing of these Wadis. The main tributaries of the Assaka Wadi (Seyad and Oum Laachar) have been planning for the spreading of flood waters. Thus, seven thresholds derivation (small dam) concrete, masonry or gabions have been constructed and are used to derivate flow rates of 15 to 30 m$^3$ /s per small dam, of a total capacity of derivation of 174 m$^3$/s (Water and Environment Ministry).

On the 7[th] of January 1985, the Province of Guelmim was hit by flooding due to overflowing of the Oum Laachar Wadi, 33mm in Guelmim center and 65 mm in Bouizakarne in 53min, at a flow rate of 1000m$^3$/s(EVICC 2011b). The importance of these events is due to the socio-economical vulnerability of this area. Unfortunately, human and economic damages for these floods are not available.

**Tata sub-catchment**

Tata province has experienced several floods. History indicates that the floods that overflow Akka River in 1995 is one of the major events (see Table 3) that impacted the zone. The damage is as following: 13 deads, 3 wounded, 4 missing and 350 families homeless; as well as destruction of 655 homes (EVICC 2011b).

Table 3: Characteristics of the 1995 flood in Tata

**Materials and methods**

Data was collected from a variety of sources (see Appendix 2), including household surveys, documents, government and ministries. The data gathered pertained to the particular indicator to be calculated for five oasean sub-catchments and five urban centers. The FVI is an indicator-based index which reflects the status of a scale's flood vulnerability. This index was determined for four components social, economic, physical, and environmental.

The flood vulnerability indicators are heterogeneous. The data for each one was collected through the technical literature from official websites and reports. There are three types of data to calculate the FVI:

1. Available data which provided by official organizations;
2. Values calculated using maps or dispersed data
3. Unavailable data was approximated using the survey;

In this paper, we used and normalized different variables (indicators) for each selected spatial scale (at urban area and sub-catchment), in a numerical index that reflects the status of a region's flood vulnerability. The used tool is the flood vulnerability index, developed in 2009 by Balica et al. The overall FVI for each scale was determined by the calculating the index from the 36 indicators for sub-catchment scale and 34 for urban scale. These indicators have been linked with the three factors of vulnerability: susceptibility, exposure and resilience. The general FVI Eq. (1) links the values of all indicators to flood vulnerability components (social, economic, physical and environmental) and factors (susceptibility, exposure and resilience), without balancing or interpolating from a series of data.

$$FVI = \frac{Exposure * Susceptibility}{Resilience} \quad (1)$$

The indicators belonging to exposure and susceptibility increase the FVI therefore they are placed at the nominator; however the indicators belonging to resilience decrease the FVI, this is why they are placed at the denominator (Dinh et al, 2012).

The calculation of each component, both in the urban and sub-catchment scale are based on the following equations (Balica & Wright 2010):

**Urban scale:**

- Social component:

$$FVI_s = \left[ \frac{P_D,\ P_{FA},\ C_H,\ P_G,\ HDI,\ C_M}{P_E,\ A/P,\ C_{PR},\ S,\ W_S,\ E_R,\ E_S} \right] \quad (2)$$

- Economic component:

$$FVI_{Ec} = \frac{I_{ND},\ C_R,\ U_M,\ I_{neq},\ U_G,\ R_T}{F_I, AmInv, D\_S_C,\ D} \quad (3)$$

- Environmental component :

$$FVI_{EN} = \frac{UG, Rainfall}{Ev, LU} \quad (4)$$

- Physical component :

$$FVI_{Ph} = \left[ \frac{T,\ C_R}{E_V/R_{ainfall}, S_C/V_{year},\ D\_L} \right] \quad (5)$$

**Sub-catchment scale**

- Social component:

$$FVI_s = \left[ \frac{P_{FA},\ R_{Pop},\ \%_{disable},\ C_m}{P_E,\ A/P,\ C_{PR},\ W_S,\ E_R,\ HDI} \right] \quad (6)$$

- Economic component:

$$FVI_{Ec} = \left[ \frac{L,\ U_M,\ I_{neq},\ U_A}{L_{EI}, F_I, AmInv,\ S_C/V_{year},\ E_{CR}} \right] \quad (7)$$

- Environmental component:

$$FVI_{En} = \left[ \frac{R_{ainfall}, D_A,\ U_G}{L_U,\ E_V,\ N_R,\ Unpop} \right] \quad (8)$$

- Physical component:

$$FVI_{Ph} = \left[ \frac{T}{E_V/R_{ainfall}, S_C/V_{year},\ D\_L} \right] \quad (9)$$

Data for calculating the FVI (and initially setting the response levels) were collected for five oasis provinces: Guelmim, Tata, Zagora, Ouarzazate and Errachidia to provide some initial testing of the model. These data were obtained from national and regional reports.

A standardization method was used for adjusting indicator values in a scale from 0 to 1 (See Table 4). The standardized formula of the FVI is as follow Eq. (10):

$$FVI_{STANDARDIZE} = \frac{FVI_{Scale}}{FVI_{Max}} \quad (10)$$

The flood vulnerability analysis was done using a detailed evaluation of the four components of flood vulnerability: social, economic, environmental and physical. These components were gathered and calculated to give the overall vulnerability.

The application of this formula for each component leads to four distinct FVI indices; $FVI_{Social}, FVI_{Economic}, FVI_{Environmental}$ and $FVI_{physical}$., which aggregates into:

$$\text{Total } FVI = \sum FVIs, FVIec, FVIen, FVIph \quad (11)$$

The FVI of each of the social, economic, environmental and physical component is computed using Eq. (1). The results of each FVI component (social, economic, environmental and physical) are summed up in Eq. (11). The FVI methodology does not require researchers to judge the relative importance of different components, i.e. they do not need to develop arbitrary weights for the indicators. The Eq. (1) links the values of all indicators to flood vulnerability components and factors (exposure, susceptibility and resilience), without weighting, as suggested by Cendrero and Fischer (1997). This is done because of different number of rating judgments which "lie behind combined weights", or interpolating.

[revised manuscript text omitted]

For the applicability of Flood Vulnerability Index and as for all methods of modeling numeric data, the FVI is associated with some points of strengths and weaknesses. The strengths of this method: the FVI allows us to gather indicators for all aspects of flood vulnerability; it allows also integrating quantitative and qualitative for different scales (sub-catchment and urban scales) in order to compare local vulnerability to floods (in four different components).

However, the major weaknesses of this index are the data collection that is much dispersed data, difficulties of access and a high cost of data collection. For our case, we have gathered, compiled, some times estimated all the necessary data from several documents and official websites (as mentioned in appendix 2, column 5).

**6. Conclusion**

The methodology used in this paper, is based on several indicators for different factors and two geographical scales, focusing on fluvial and urban floods. Various indicators were taken into account to assess flood vulnerability. The Flood vulnerability Index was use for these case studies at two scales (sub-catchment scale and area scale). Lot of data was needed to estimate the flood vulnerability indicators for each of the two areas. An accurate assessment of flood vulnerability is difficult, due to the lack of official necessary data. However, in order to complete this evaluation, more data were collected via social questionnaires, official documents and websites (see Appendix 2).

The FVI can show readily implicit and readily communicated results that can help decision- makers in identifying the most effective measures to be taken. Uncertainty is not removed, but is integrated into the assessment. On the other hand the complexity of FVI methodology is also a negative point, since it takes a long time and good knowledge of the area and the system behind the FVI to be able to implement it.

Social indicators are difficult to quantify. On the other hand, such a parametric method can give a basic way of characterizing what in reality is an intricate system. Such results will help to give an indication of whether a system is resilient, susceptible or exposed to flooding risks and help identify which measures would reap the best return on investment under a changing climate and population and development expansion.

**At urban scale,** the Errachidia (0.09) is small vulnerable, while the four other cities (Zagora (0.47), Tata (0.28), Ouarzazate (0.26), and Guelmim (0.25)) are vulnerable to floods. Regarding the **sub-catchment scale** the Errachidia (0.55), Guelmim (0.54), and UDV (0.5), have very similar values. These values lead to classify the three sub-catchments as high vulnerable to floods. Regarding the two other sub-catchments, can be seen that MDV (0.37) is vulnerable, and Tata (0.073) has small vulnerability to flood. Comparing the whole components, the Errachidia is the most vulnerable, with the exception of the physical component.

Concerning the applicability of Flood Vulnerability Index methodology can be summarized as follows:

- Vulnerability can be reflected by exposure, susceptibility and resilience factors;
- The sub-catchment and urban systems can be damaged regarding four different components estimated in the result section.
- The FVI is adaptable to different uses in the Moroccan pre-Saharan region.
- This tool allows to identify the risks and the management methods to assess flood vulnerability;
- Find out the priority components and sectors of flood vulnerability in order to take urgent measures
- The FVI is applicable in sub-catchment and urban area scales in pre-Saharan region.
- Finally, the proposed methodology to calculate a FVI provides an approach to quantify how much floods are affecting, or can affect a sub-catchment or an urban area in pre-Saharan regions.

[revised manuscript text omitted]

---

## Referee Comment (RC2) · Anonymous Referee #2 · 13 Jun 2016

The paper "Analysis of applicability of flood vulnerability index in pre-saharan region.." presents the application of an establish method of computing flood vulnerability indexes for a particular catchment. Though the method is not new its applicability in pre-saharan region is interesting to be looked at. What I did missed however is that the paper presents a very simple step by step application of the method with no new additions, nor testing its applicability in the region. It is a simple gathering of data and crunching it in a formula. I do not know what is a difference between the results presented in this paper and a technical report evaluating vulnerability to floods in the region.

Apart from the fact that the English narration of the paper is difficult to read, the

manuscript needs a lot of work before it can be brought to a publishable level. I would suggest the authors to re-write the paper with a research component in it, to have a new added value to the tested methodology. The authors may follow their own title suggestion by making a wider analysis of the method, referring to its applicability, not to its application to a particular case study. I would expect that authors will look what are the conditions in which such an index is applicable in pre-saharan region, where preparedness for floods is not so spread. What should be the indicators that would be more important than others, or maybe what are the indicators that can be left out.

Discussion is very short and quite vague, not focusing on the findings from the application of the method. Also conclusion part is very vague, not focusing enough on the region itself, as promised in the title of the article.

I have minor comments on the text that I could add after the manuscript revision of the concept has been done.